# Knowledge Extraction and Discovery about Web System Based on the Benchmark Application of Online Stock Trading System

**DOI:** 10.3390/s23042274

**Published:** 2023-02-17

**Authors:** Marcin Borowiec, Rafał Piszko, Tomasz Rak

**Affiliations:** Department of Computer and Control Engineering, Rzeszow University of Technology, Powstancow Warszawy 12, 35-959 Rzeszow, Poland

**Keywords:** experimental analysis, workload characterization, web client classification, web benchmark

## Abstract

Predicting workload characteristics could help web systems achieve elastic scaling and reliability by optimizing servers’ configuration and ensuring Quality of Service, such as increasing or decreasing used resources. However, a successful analysis using a simulation model and recognition and prediction of the behavior of the client presents a challenging task. Furthermore, the network traffic characteristic is a subject of frequent changes in modern web systems and the huge content of system logs makes it a difficult area for data mining research. In this work, we investigate prepared trace contents that are obtained from the benchmark of the web system. The article proposes traffic classification on the web system that is used to find the behavior of client classes. We present a case study involving workload analysis of an online stock trading application that is run in the cloud, and that processes requests from the designed generator. The results show that the proposed analysis could help us better understand the requests scenario and select the values of system and application parameters. Our work is useful for practitioners and researchers of log analysis to enhance service reliability.

## 1. Introduction

Measurement technology has evolved into wearable hardware devices, but we also can use observers as parts of software that transmit monitored values of system variables and collect them in databases. Usually, these captured data from production servers are technically difficult or relatively expensive. Furthermore, tests of the production system are not possible or are carried out too late. However, in many cases, it is necessary to test the hardware and software environment already at the early stages of its preparation. Analysis of the Internet system is a complex and time-consuming task and requires appropriate preparation on both sides—the software and the hardware. Logs report information, which are crucial to diagnose the root cause of complex problems. Administrators of most user-facing systems depend on periodic log data to get the status of production applications. Experimental environments make it possible to create an ecosystem for collecting and processing data about a specific environment so it could be monitored, managed, and controlled more easily and efficiently. Therefore log data is an essential and valuable resource for online service systems.

In this article, an experimental environment will be presented, as a case study, based on a container structure. Furthermore, a web application (AP1) was developed, which their task is to perform the appropriate tasks algorithm. Modern web systems provide multiple services that are deployed through complex technologies. Thus, this approach, on the software and server side, is based on the latest programming technologies and containers running in the cloud. The proposed application (AP1) is a new concept based on the DayTrader Java EE application [1] originally developed by IBM as the Trade Performance Benchmark Sample. DayTrader is a benchmark application built around the paradigm of an Online Stock Trading System (OSTS). In order to automate the testing process of the prepared application, an additional application was constructed, which is an automatic client (AP2), that generates players and performs requests on the OSTS. The combination of these two applications allowes for the preparation of a benchmark that was used to analyze web requests. The process of playing on the OSTS was analyzed with predetermined query scenarios (A1, A2, A3). The OSTS tasks include receiving and then processing purchase and sale offers while measuring the duration of individual operations, conducting transactions, i.e., executing previously created offers, and measuring the CPU and RAM consumption of each container.

Based on this approach, it was possible to analyze the behavior of the OSTS and the requests processing during increased load. Data was obtained in tests for various test parameters on two different hardware architectures AR1 and AR2 and with a different number of *R* docker replicas because the benchmark program was placed in a prepared container environment. This solution uses the system architecture where communication is mediated by a message queuing server.

Modern system development and operations rely on monitoring and understanding systems behaviour in a production. Behavior analysis was performed and used for client traffic classification. It was possible to indicate customer traffic based on the system parameters obtained and the application processing. Traditional analysis and prediction methods in cloud computing provide unidimensional output. However, the unidimensional output cannot capture the relationship between multiple dimensions, which results in limited information and inaccurate results. The precise determination of customer behavior is very difficult, but with the use of multidimensional hardware, software factors and the defining of trends of clients’ behavior it has been successful.

The rest of this article is organized as follows. We discuss related work and introduce our previous models in Section 2. Section 3 presents our solution based on hardware and software elements. This section contains mechanisms describing the system, the operation of the OSTS, and the game algorithms implemented by the generator. In Section 4, we evaluate the usefulness of our benchmark for analysis in the domain of the web system. Finally, Section 5 presents the conclusions and future work.

## 2. Related Works

In recent years, novel applications have emerged and it benefited from automated log-file analysis, for example, real-time monitoring of system health, understanding user behavior, and extracting domain knowledge. In [2,3], we can find a systematic review of recent literature (covering the period between 2000 and June 2021) related to automated log analysis. Application logs record the behavior of a system during its runtime, and their analysis can provide useful information. Log data is used in anomaly detection, root analysis, behavior analysis, and other applications. In this section, we discuss related work on this topic for web system structures. We divided related work into two parts software and hardware. Some articles present several methods to design new and improve existing web systems that, even within unpredictable load variations, have to satisfy performance requirements [4]. Reliability testing is a significant method to ensure the reliability and quality of systems. The proposition in [5] taxonomy to organise works focusing on the prediction of failures could help in the context of Web structures performance. This taxonomy classifies related work along the dimensions of the prediction target (e.g., anomaly detection, performance prediction, or failure prediction), the time horizon (e.g., detection or prediction, online or offline application), and the applied modeling type (e.g., time series forecasting, machine learning, or queueing theory). We were able to find works for understanding workloads and modeling their performance is important for optimizing systems and services. The main models were presented in [6,7] uses Petri Nets. Article [8] presents a method for setting the input parameters of a production system. In [9], authors try to understand and model storage workload performance. They analyzed over 250 traces across 5 different workload families using 20 widely used distributions. Publications [10,11] use stochastic formalisms for the performance engineering of a web system and compare their own models with the performance of the production system. We based on strategies and techniques that could be used in practice to derive the values of common metrics, including event-driven, tracing, sampling, and indirect measurement proposed in [12]. Furthermore, some of them could be applied generally to other types of metrics.

The load generators [13,14] that define web workloads imitate the behavior of thousands of concurrent users in a web browser. Existing generators mostly use different distributions for representing the time between requests (client think time) [15].

We found many old benchmarks, but they are all based on old types of system. WebTP [16] is a benchmark that measures the performance of a web information subsystem. In [17], we can find old techniques of performance testing and various diagnostic tools to implement testing. In recent years, several new tools and methodologies have been used to evaluate and measure the quality of web systems. For example, we could find [18] checking the conformance with respect to the requirements (compatibility testing). In this context, one challenge for analysis is how to execute multiple test cases, in a correct and efficient way, that may cover several environments and functionalities of the tested applications while reducing the consumed resources and time. The existing approaches suffer from several limitations when deploying them in practice [19]: inability to deal with various logs and complex log abnormal patterns, poor interpretability, and lack of domain knowledge. Logfile anomaly detection is vital for service reliability engineering. Paper [19] proposes a generic log anomaly detection system based on ensemble learning. They conduced an empirical study and an experimental study based on large-scale real-world data. In [20], the authors conducted a comprehensive study on log analysis in Microsoft. This article uncovers the real needs of industrial practitioners and the unnoticed, yet the significant gap between industry and academia. Debnath [21] presents a real-time log analysis system that automates the process of detecting anomalies in logs. This system runs at the core of a commercial log analysis solution that handles millions of logs generated from large-scale industrial environments. In [22], the authors offered a method for conceptualizing and developing a real-time log acquisition, analysis, visualization, and correlation setup for tracking and identifying the main security events.

New technologies have not only offered new opportunities but also have posed challenges to hardware and software reliability technology. In [23], the technologies of software reliability testing were analyzed, including reliability modeling, test case generation, reliability evaluation, testing criteria, and testing methods. Proposed in [24] framework can predict the resource utilization of physical machines. This framework consists of two parts: a noise reduction algorithm and a neural network. Davila-Nicanor in paper [25], presents a process to estimate test case prioritization on Web systems. The results become a guide to establish test coverage through the knowledge of the most critical paths and components of the system. The newest and the most common proposed hardware behavior predictions are based on machine learning techniques [26,27]. In [26], proposes a novel Prediction mOdel based on SequentIal paTtern mINinG (POSITING) that considers the correlation between different resources and extracts behavioral patterns. Based on the extracted patterns and the recent behavior of the application, the future demand for resources is predicted. Reliability, availability, and maintainability aspects are critical for an engineering design and were investigated in [28]. These aspects concern a system’s sustained capability throughout its useful life. The authors in [29] provided a methodology that results in the successful integration of Reliability, Availability, and Maintainability with Model-Based Systems Engineering that can be used during the early phases of design.

Paper [30] analyzes the time-sharing system and the network connection, by exploring internal computer processors. Some works [31] are related to the detection of errors in the process of static software analysis. Said et al. [32] presented a straggler identification model for distributed environments using machine learning. This model uses several parameters extracted by the execution of various types and large-scale jobs. In the paper [33], the authors presented the study of workload prediction in the cloud environment.

In many cases, traffic analysis and classification of web system requests only include models of native architecture. All use experiments to verify the proposed classifications. However, we were unable to find an approach based on a container architecture. Furthermore, we could not find an approach applicable to up-to-date software web framework tools. Some authors present tools for the run-time verification of quantitative specifications applications. The PSTMonitor from [34] is the detection of executions that deviate from the expected probabilistic behavior. CaT [35] is a nonintrusive content-aware tracking and analysis framework. CaT can improve the analysis of distributed systems. The paper [36] presents LogFlow, a tool to help human operators in the analysis of logs by automatically constructing graphs of correlations between log entries. The core of LogFlow is an interpretable predictive model based on a Recurrent Neural Network.

## 3. Web System Hardware and Software Architecture

Next, we will describe the experiment environment: hardware (AR1 and AR2) and software (APP1 and APP2). We will use them for evaluation in this case study.

### 3.1. Hardware Architecture

In this subsection, we describe the hardware test environment that we used for the analysis. We deployed our system on the Docker container platform. The back-end servers contain a logic tier and a database. To reflect a realistic container setup (Docker engine in Swarm mode), we deployed the application APP1 as a benchmark inside a container with Linux OS. The load generator for requests APP2 was situated in another container. Depending on the test, we used a different number of containers. We used a pool of containers running on a cloud that had: AR1 (8 CPU and 20 GB RAM) and AR2 (12 CPU and 30 GB RAM). A separate internal network connected all containers. We ran only the programs expected for tests. All parts of the system were implemented based on a PIONIER cloud environment (https://cloud.pionier.net.pl/, accessed on 18 December 2022) with Intel Xeon E312xx (Sandy Bridge) processors. The sample memory size was selected experimentally for request processing performance lower than 100% in all cases.

### 3.2. Software Architecture

All analyses require a set of input parameters. The concrete set of input parameters differs depending on the architecture of the underlying system and the concrete test. We carried out several test runs. In our experiments, we used an implementation based on the Java Spring Boot framework (APP1— https://github.com/raphau-soft/stock-backend-swarm, accessed on 18 December 2022 and APP2—https://github.com/raphau-soft/traffic-backend-swarm, accessed on 18 December 2022), Flyway library, MySQL open-source database management system, and RabbitMQ message broker software.

#### 3.2.1. Containers Configuration

Docker is an open platform for building, delivering, and running applications. It allows you to decouple the application from the infrastructure so that you can deliver software quickly. It allows you to significantly reduce the delay between writing code and running it in a production environment. Container use controlled portions of the host operating system resources. Docker is only one layer between the real CPU and the web application. Applications share the same operating system kernel in a highly managed way.

Using the Dockerfile files, we could build our container with the written application and then run it. Such a container has everything needed to run and at the same time is as small as possible. In addition, Docker allows us to define the structure of the entire system consisting of various applications by creating and running the file *docker-compose.yml* (https://github.com/raphau-soft/stock-swarm/blob/master/docker-compose.yml, accessed on 18 December 2022). Powerful Docker mechanism called the Docker Swarm, allows us to run a certain number of replicas of a given container. Orchestration is an advanced DevOps tool that allows accurate and automatic management of entire deployments from a single tool. The main advantages of orchestration are allowing a single person to monitor and manage hundreds of systems and automation. Orchestration allows very easy scaling of services. This greatly simplifies application load management. Please note that the Docker Swarm supports load balancing, due to which traffic and external load are shared between all replicas of a given service.

Each server runs in its container. The Docker tool allowed creation of a system of containers that could communicate with each other. As shown in the figure (Figure 1), the system consists of two databases, a query queue, the APP2 traffic generator, and any number of APP1 OSTS replicas. Docker has been run in Swarm mode, so any container can be created with a given number of replicas.

We deploy a web application in a container in the cloud.

#### 3.2.2. RabbitMQ

The use of RabbitMQ allows for asynchronous processing of sent requests in both directions, so applications do not wait for each other. Division was introduced into producers, i.e., those who generate messages and consumers who receive and process these messages. The table (Table 1) shows the queues and information on which application in the case of a particular queue is a consumer *C* and which is a producer *P*.

### 3.3. Benchmark

We need a dedicated and controllable benchmark. Benchmark is a web application that simulates the operation. It is patterned on the IBM DayTrader Benchmark designed to be representative of typical Internet commerce applications. The benchmark consists of a database and a set of Application Programming Interface endpoints. It is also directly related to the stock exchange system (OSTS). The tests include more or less complex activities, where the system may only carry out one type of operation (buying/selling). We use a load generator that generates a defined system traffic. APP2 emulates user behavior and performs requests to stress a web API. During the experiment runs we obtain measurements of the response times of individual requests and resource utilization. Its logs contain the response time of every request. The APP2 runs on a separate container.

Individual queues perform the following tasks and transmit:buy−offer−request—creates purchase offers,sell−offer−request—creates sales offers,company−request—creates companies,test−details−response—sends a response with time data of operations,cpu−data−response—the measurement of CPU and RAM usage,user−data−request—requests for user data,stock−data−request—requests for company data,user−data−response—a response with user data,stock−data−response—a response with company data,register−request—user registration data,register−response—registration confirmation.

The processing of bids and offers is the basic operation of the APP1 exchange. The scheme is common for both buy and sell offers. In the case of a buy offer, the player’s portfolio is updated, and in the case of a sell offer, the shares held by the player are updated. First, the OSTS receives a buy or sell message from the queueing server. Then, it downloads from the database such data as the user and the company to which the offer applies and, in the case of a sale offer, the shares that are the subject of this offer. The data are updated and saved in the database together with the created offer. In these operations, the code responsible for measuring the overall processing time and the time of operations on the database, i.e., when downloading and saving data, is intertwined. These measurements are then sent to the queuing server and saved on the traffic generator side to the APP2 generator database. The time from sending a message to receiving it is largely dependent on the status of the queue; therefore, an additional parameter in the response sent to the traffic generator is the current number of messages in the queue before sending the response with measurements.

### 3.4. Transactions Flow

The entire scheme for conducting transactions on the OSTS is presented in the figure (Figure 2). In the first step, the application downloads a list of all companies in the database. The current buy and sell offers are then retrieved for each company and then sorted so that the highest-priced buy bids and the lowest-priced sell bids are considered first. The next step is pairing offers with each other and carrying out transactions, i.e., the buyer loses money and gains shares, and the seller vice versa. This happens as long as there are offers in the database with which further transactions could be made. When these offers are no longer available, the algorithm moves on to the next company.

## 4. Analysis of Requests Traffic

One of the ways to guarantee high-quality applications is through testing. Testing is an important aspect of every software and hardware development process that companies rely on to elevate all their products to a standardized set of reliable software applications while ensuring that all client specifications are met. Due to the lack of traces from the real container system, instead we use an application benchmark for the prepared scenario. All analyses require a set of input parameters. The concrete set of input parameters differs depending on the underlying system architecture and particular test.

This section describes the test setup used to obtain the measurement traces. The benchmark allows us to use measurements for performance parameters. A set of experiments was conducted and the results were analyzed. We collected observations of the arrival times, execution times of individual requests, and average CPU utilization during each experiment run. The benchmark execution times were measured.

The tests have been grouped into 4 characteristic groups {K1,K2,K3,K4} examining different characteristics:K1 group examining the impact of the number of replicas (repl).K2 group examining the impact of time between transaction execution (trans).K3 group examining the impact of time between player requests (req).K4 group investigating the impact of scenarious used by players (A1, A2, A3).

This study also has taken into account the impact of the physical factor, i.e., the performance of the hardware itself on which the tested application will be launched. The tests were carried out on two different servers with different architectures. The first architecture has 8 processors and 20 GB of RAM (AR1), while the second has 12 processors and 30 GB of memory (AR2).

### 4.1. Game Strategies

Each generated user (player) has its specific class and according to that it takes actions on the OSTS. Before the test, parameters are given that determine how many players should be launched for a given algorithm. The results obtained depend on the values of the given parameters.

The first algorithm is “Buy and sell until resources are used”, hereinafter referred to as A1. Its operation diagram is shown in the figure (Figure 3). The second algorithm is “Buy and sell alternately”, hereinafter referred to as A2. The scheme of its operation is shown in the figure (Figure 4). The third algorithm is “Just browse”, hereinafter referred to as A3. Its diagram is shown in the figure (Figure 5). The difference between the A1 and A2 algorithms is that the A2 algorithm does not run out of resources, i.e., it adds one buy then one sell, while the A1 algorithm adds a buy offer in a loop until the player’s resources run out, then adds sales offers until the player’s resources run out as well. The A3 algorithm does not affect the expansion of data in the database of the APP1 application because these are read-only operations.

### 4.2. Tests Configuration

Each of the 18 tests (with 4 traces per test) presented in the table (Table 2) was also performed for different time ranges: 1 h, 3 h, 6 h, 9 h, and 12 h (360 logs per architecture):K1={Test1,Test2,Test3,Test4}—is characterized by a change in the number of replicas,K2={Test5,Test6,Test7,Test8,Test9}—is characterized by a different times between transactions,K3={Test10,Test11,Test12,Test13}—is characterized by a different times between requests,K4={Test14,Test15,Test16,Test17,Test18}—is characterized by a different number of players realising a given algorithm (strategy).

After the set time limit had elapsed, it was possible to download the data collected by the benchmark:logs of queries made by the player, e.g., issuing an offer,logs of consumption parameters of APP1 replicas—CPU and RAM memory,logs regarding the number of issued purchase/sale offers,CPU and RAM memory usage logs for APP2.

### 4.3. Experiment Results

In this subsection, we present the results obtained from the benchmark tests. In many production situations, direct measurement of resource demands is not feasible. Benchmark testing is a normal part of the application development life cycle and is performed on a system to determine performance. Finally, we describe the conducted experiments in detail and present the results obtained from them. During the experiments, we monitored each container as well as every request. Benchmark tests are based on repeatable environments. We carried out experiments in realistic environments to obtain measurement traces.

#### 4.3.1. Impact of the Number of Replicas of the Stock Exchange Application on Performance

The OSTS is scalable; i.e., it allows the determination of the number of replicas of the application to increase the efficiency and responsiveness of the entire system. The desired feature of scalability in the OSTS it has been implemented with the use of containerization software and the Swarm mode built into it, which implements this mechanism. As you can easily guess, increasing the number of replicas of the OSTS should allow the server to better use the available resources and the operation of the application itself. In the case of the APP1 architecture, the difference in processor utilization after increasing the number of replicas from 2 to 6 increases by a maximum of a few percent, in the K1 test group (Figure 6), which allows for a slightly larger number of requests during a given time limit *T*, e.g., offers to buy and sell shares (Figure 7 and Figure 8).

#### 4.3.2. Impact of Architecture on the Stock Exchange Application Operation

The obvious fact is that the operation of the system in terms of its performance depends on the platform on which it is running. Carrying out system load tests in a test environment allows us to answer the question of how the application will behave, e.g., in the case of a very high load, and whether the tested architecture has sufficient resources to handle all incoming requests. This is a very important consideration when working on more sensitive systems that need to be available 24/7 with downtime kept to a bare minimum. Both tested hardware architectures, AR1 and AR2, meet the requirements of all test scenarios, i.e., they allow for their trouble-free completion within the desired time limit, and there are no complications related to the lack of hardware resources. As a consequence, the generated logs are complete. The mixed K4 test ($A1\text{\_}100,A2_{1}00,A3\text{\_}100$) simulates more similar real conditions (each player performs different actions and works according to the different scheme) to some extent represents the approximate load of the real system. According to the chart (Figure 9), we could decide which architecture solution we are going to use. By choosing the AR1 solution, we will meet the demand for server resources from clients, but for more future-proffing, this solution may not be enough, as the popularity of the service increases (peak load is a maximum of 75%). An alternative is the AR2 architecture, which will provide a greater reserve of computing power (peak load lower by 15%), and thanks to the application scalability mechanism, it is possible to change the server’s hardware configuration. Similarly, in the case of RAM memory, by monitoring current consumption, we could determine whether its level is sufficient (Figure 10).

#### 4.3.3. Impact of the Type of User Requests on the Stock Exchange Application Load

The characteristics of the system load depend primarily on the type of requests processed by the system-actions performed at the moment by the players. As you can see in the figure (Figure 11), the time course experiment data series of most tests is sinusoidal when players use the strategies A1 and A2 (buying and selling stocks). This is because an algorithm has been implemented in the OSTS that handles transactions every $T_{T}$ interval, due to which there is a break between generated requests. Only in the scenario of 200 concurrent A3_200 players, where users only view offers (A3 strategy), the load is relatively constant and has lower load-amplitude fluctuations in the graph. In summary, the algorithm plays a key role in the load on the OSTS and stresses the system the most when it performs its task. The transaction algorithm is discussed in Section 3.4.

#### 4.3.4. The Influence of the Transaction Algorithm on the Operation of the Stock Exchange Application

The transaction algorithm executes transactions, i.e., it combines buy and sell offers based on the queuing mechanism (First In First Out) and checks a number of conditions that have to be met for the exchange of shares to take place. An important parameter of the OSTS is to specify the time interval ($T_{T}$) in which successive batches of buy/sell offers will be processed using this algorithm. Setting it at the right time can have a positive effect on the processing of offers and has the correspondingly benefitial effect on load-balancing of the server. Time periods of 1 to 5 min delay between processing have been tested.

For the AR1 architecture (Figure 12), the most appropriate delay between transactions K2 was the time lower than 60 [s]. Execution of requests on a regular basis results in lower consumption of server resources resulting in the lack of a long queue of buy/sell offers (Figure 13). This results in a smaller number of processed offers within $T_{T}$ (Figure 14), leading to less impact on server resources (Figure 12).

Considering the second AR2 architecture of the above-mentioned relationships, we also have observed the same behavior. It is interesting that for $T_{T}$ = 60 [s], AR2 generates a linear load, except for tests with a higher delay (Figure 15). This phenomenon was also observed in longer tests. To sum up, the appropriate setting of the $T_{T}$ delay parameter for the transaction algorithm positively affects the operation of the OSTS; however, the correct operation of the system should be verified by examining its time logs. The too-low and too-high values of these delay cause problems with the functioning of the application, which has been checked.

#### 4.3.5. Time between Requests

The last relationship discovered in the log analysis process was based on the K3 scenario group analyzing the impact of the $T_{R}$ parameter (time between player queries) on the final logs of the OSTS. The smaller the time interval (think time), the more the system is loaded with player requests and, therefore, requires more resources. The think time adds delay in between requests from generated clients [4]. Of course, it is unrealistic for each player to perform various actions on the website in such short time intervals, but the simulation clearly shows the high impact of this parameter on the test results (Figure 16). In addition, another dependence was verified, that could be read from the charts—the CPU consumption is the same on each container.

## 5. Conclusions

In the era of intelligent systems, the performance, reliability, and safety of systems have received significant attention. Predicting all aspects of the system during the design phase allows developers to avoid potential design problems, which could otherwise result in reconstructing an entire system when discovered at later stages of the system or software development life cycle. To ensure the desired level of reliability, software engineering provides a plethora of methods, techniques, and tools for measuring, modeling, and evaluating the properties of systems.

In this article, a novel design concept is presented as a case study of a container-based web system in the cloud. The aim of doing so is to demonstrate the key changes in systems design activities to address reliability and related performance. It is worth noting that web system modeling is helpful in the context of both correlated efficiency growth and behavior recognition.

We designed and implemented a tool for an expanded analysis based on performance parameters from logs. We conducted a long-term study of the Online Stock Trading System (OSTS). We applied approaches for analysis using the system logs of the benchmark for different workload scenarios and different hardware/software parameters. Along with measurements, we presented some conclusions about the characteristics of requests. Lastly, we evaluated these values in relation to the suitability for recognizing the request stream.

A benchmark was prepared based on a container structure running in the cloud, which consists of elements such as exchange replicas, traffic generator, queuing server, OSTS database, and a traffic generator database. The task of the generator was to run a test consisting of simulating a certain number of players of the selected class, that then sends queries to the OSTS via the queuing server. During this test, data on query processing time, CPU, and RAM usage were collected for each container. The next step was to analyze the obtained data. We identified the following obvious benefits: CPU usage is the same on every replica of the exchange, the more requests in the queue, the longer the processing time, A3 algorithm generates constant CPU usage, CPU usage is much lower on AR2 architecture, AR2 architecture processes more queries than the AR1 architecture, with a short break between requests, the AR1 architecture is unable to send more requests quickly, and a short transaction time keeps request-processing times low.

By examining logs, it is possible to gain valuable insights into how the application is functioning and the manner in which it is being used by users. This information can be examined to identify any issues that need to be addressed, as well as to optimize the performance of the application. Logs can be also used to identify elements of the application that can be further optimized. Additionally, log analysis can be useful for identifying trends in the use of the application, allowing for a deeper understanding of user preferences. It remains the choice of system developers to determine how detailed the logs generated by the system will be. Consequently, this can greatly enrich the results of ongoing research and provide valuable information.

The main contribution of this paper is the discussion of the issues (e.g., business models) involved in creating benchmark specifications for up-to-date web systems. The presented results show that there are many possible links between web requests or web traffic and the production system. The proposed benchmark can help by providing guidelines for the construction of a container-based cloud-based web production system. A holistic view of the research effort on logging practices and automated log analysis is key to providing directions and disseminating the state of the art for technology transfer.

This work also highlights several possible future research directions. In future work, the limitations to expanding the presented results should be discussed. The main research topic should center on the use of many request classes in one scenario, which will bring the model closer to reality. Another step could be to check the players’ behavior in the second test scenario and its influence on the response time. Building a performance model with high generalization and providing more interpretable reconstruction results for the data-driven model are important tasks for our future research. Finally, we consider proposing a method of discovering anomalies in web systems and application logs based on user behavior.

## Figures and Tables

**Figure 1 sensors-23-02274-f001:**
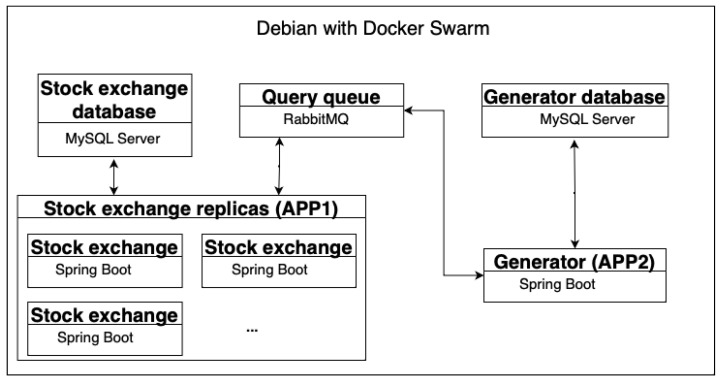
System schema.

**Figure 2 sensors-23-02274-f002:**
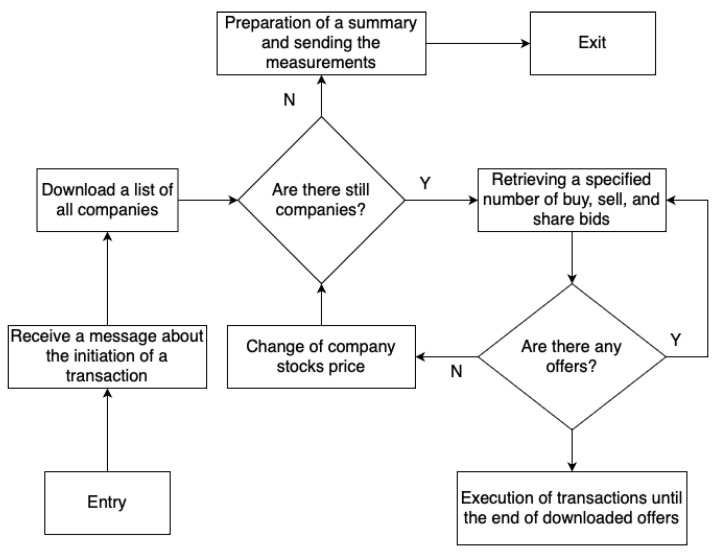
Transaction processing scheme.

**Figure 3 sensors-23-02274-f003:**
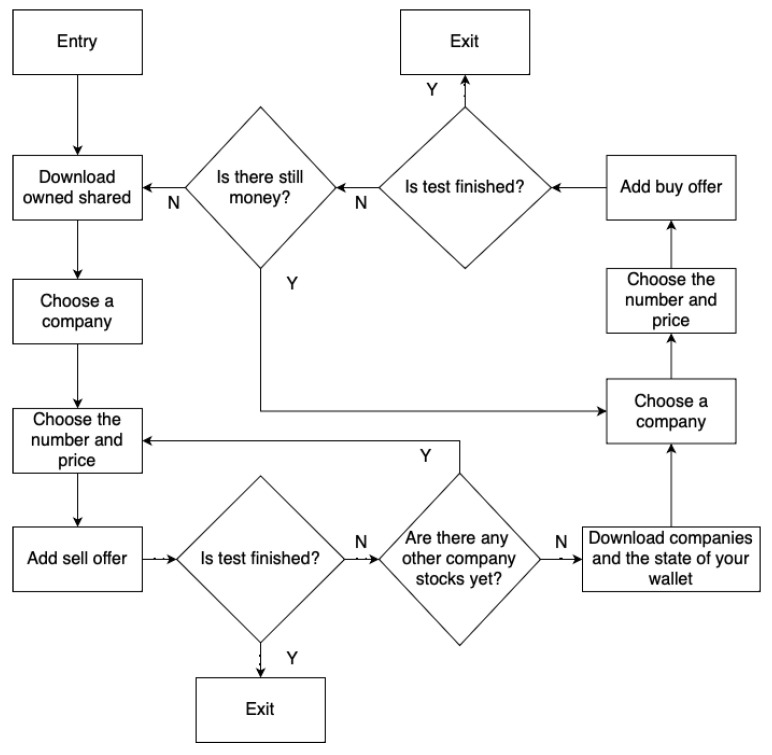
A1 schema.

**Figure 4 sensors-23-02274-f004:**
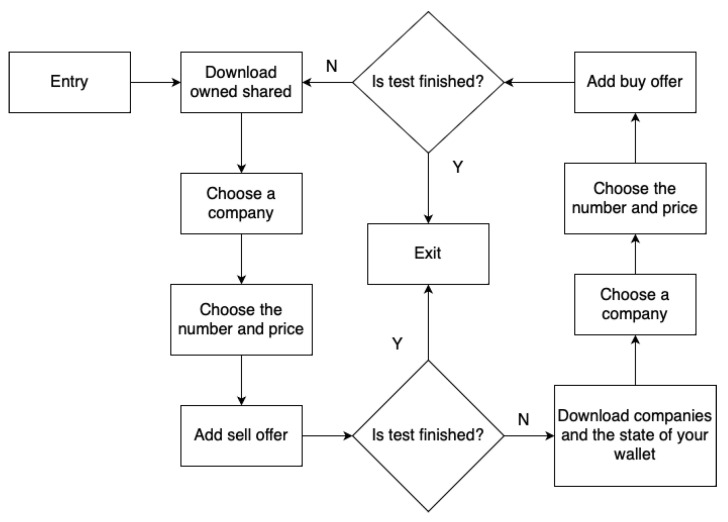
A2 schema.

**Figure 5 sensors-23-02274-f005:**
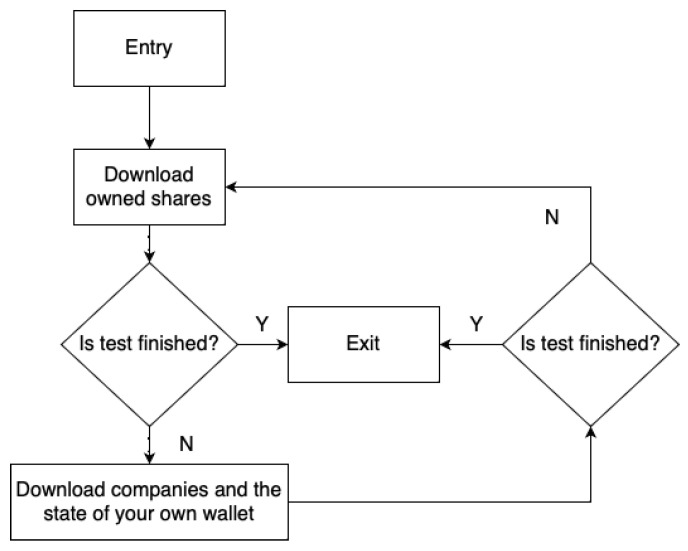
A3 schema.

**Figure 6 sensors-23-02274-f006:**
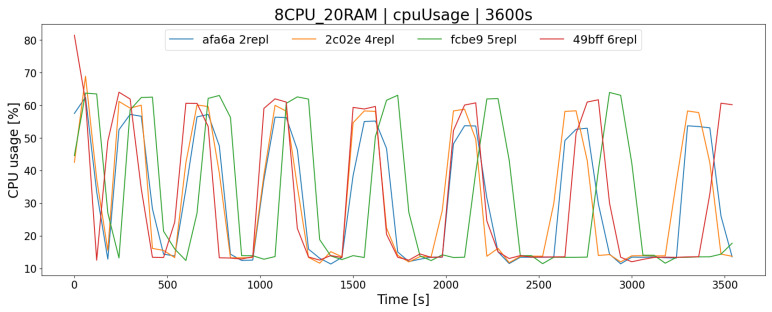
Percentage of CPU usage in the AR1 architecture depending on the number of replicas of the stock exchange application (repl test group).

**Figure 7 sensors-23-02274-f007:**
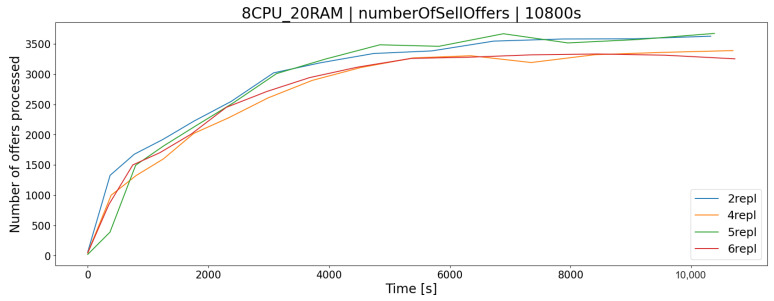
The number of processed stock sales offers in relation to the number of stock exchange application replicas.

**Figure 8 sensors-23-02274-f008:**
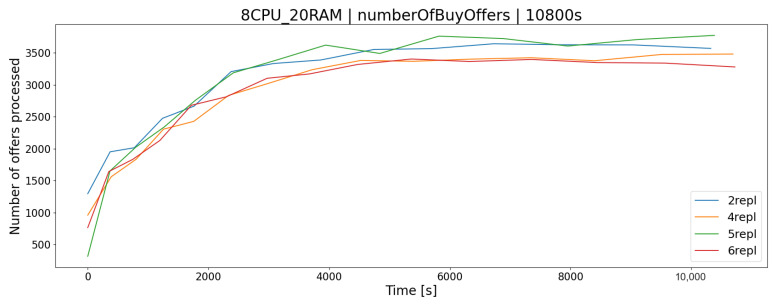
The number of processed stock buy offers in relation to the number of stock exchange application replicas.

**Figure 9 sensors-23-02274-f009:**
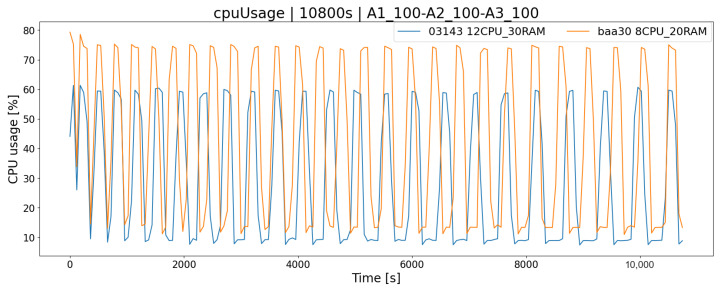
CPU usage of two architectures-mixed scenario.

**Figure 10 sensors-23-02274-f010:**
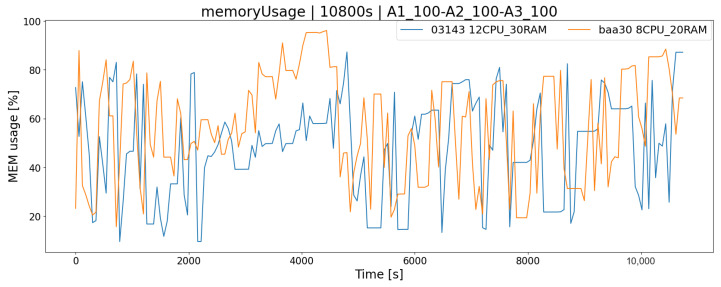
Memory consumption of two architectures-mixed scenario.

**Figure 11 sensors-23-02274-f011:**
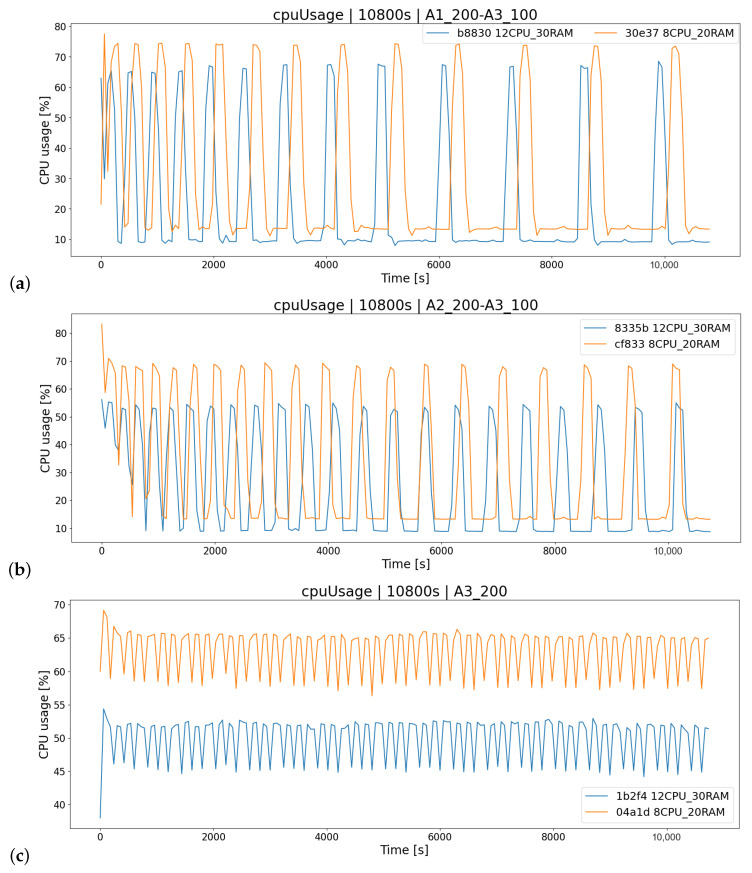
Measuring CPU usage of two architectures-test scenarios K4: (**a**) algorithm A1 with 200 players and algorithm A3 with 100 players, (**b**) algorithm A2 with 200 players and algorithm A3 with 100 players, (**c**) algorithm A3 with 200 players.

**Figure 12 sensors-23-02274-f012:**
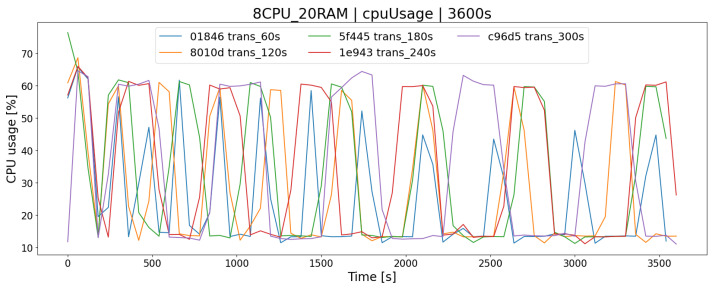
Measuring CPU usage for the latency parameter of the transaction algorithm (AR1 architecture).

**Figure 13 sensors-23-02274-f013:**
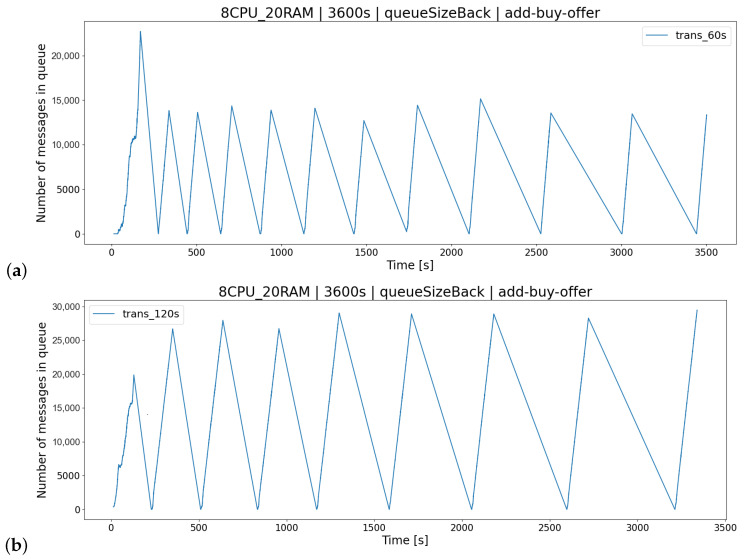
An example of an increase in pending requests in the queue for adding purchase offers for: (**a**) $T_{T}$ = 60 [s], (**b**) $T_{T}$ = 120 [s].

**Figure 14 sensors-23-02274-f014:**
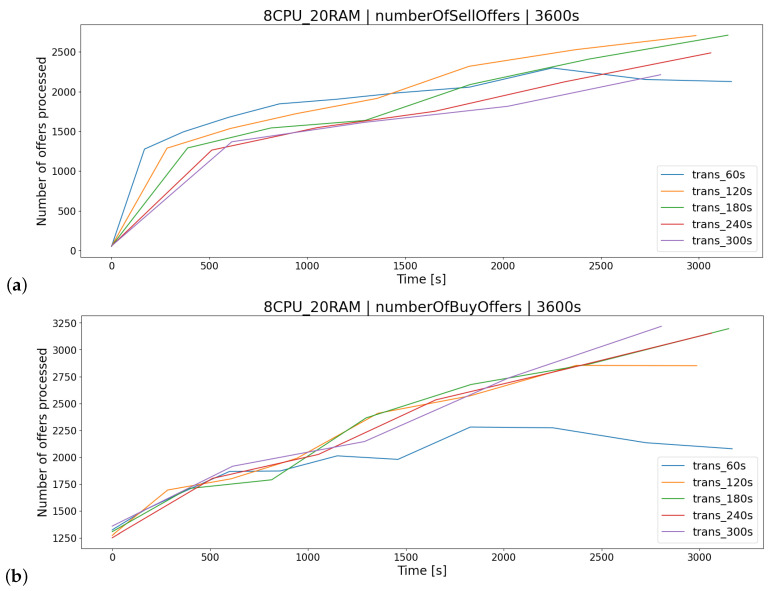
Number of sell (**a**) and buy (**b**) offers carried out during the one-hour test for the test group examining the transaction delay parameter (trans).

**Figure 15 sensors-23-02274-f015:**
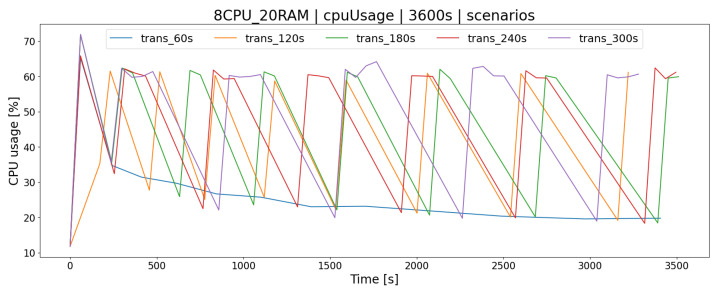
Measuring the CPU usage of the traffic generator (APP2) of the stock exchange application.

**Figure 16 sensors-23-02274-f016:**
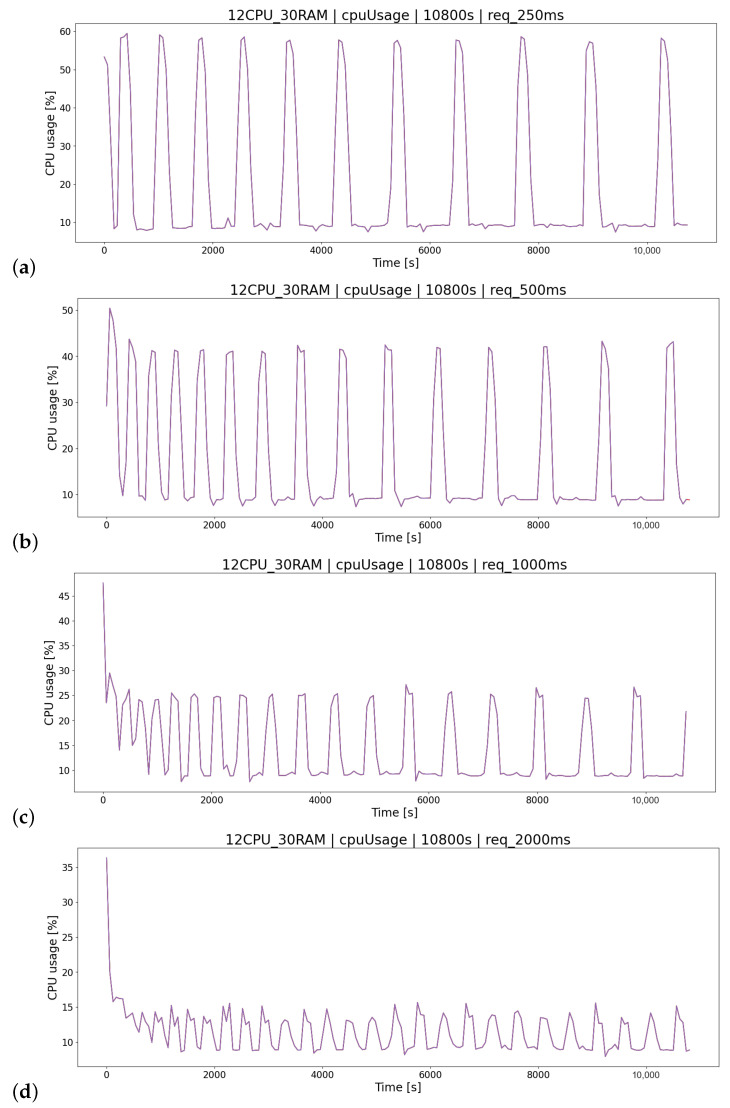
CPU consumption measurement for scenarios testing query delay parameter: (**a**) $T_{R}=250\left[ms\right]$, (**b**) $T_{R}=500\left[ms\right]$, (**c**) $T_{R}=1000\left[ms\right]$, (**d**) $T_{R}=2000\left[ms\right]$.

**Table 1 sensors-23-02274-t001:** Producers and consumers.

Queue	APP1	APP2
buy−offer−request	*C*	*P*
sell−offer−request	*C*	*P*
company−request	*C*	*P*
test−details−response	*P*	*C*
cpu−data−response	*P*	*C*
user−data−request	*C*	*P*
stock−data−request	*C*	*P*
user−data−response	*P*	*C*
stock−data−response	*P*	*C*
register−request	*C*	*P*
register−response	*P*	*C*
trade−request	*C*	*P*
trade−response	*P*	*C*

**Table 2 sensors-23-02274-t002:** Test scenarios for 8CPU_20RAM (AR1) and 12CPU_30RAM (AR2) architectures for 4 groups ${}^{a}$.

Test Number	Test Name	T [s]	R	S1	S2	S3	$T_{R}$ [ms]	$T_{T}$ [s]
Test1	5repl	3600	5	200	0	0	500	180
Test2	2repl		2	200	0	0	500	180
Test3	4repl	10,800	4	200	0	0	500	180
Test4	6repl		6	200	0	0	500	180
Test5	trans_60s	21,600	5	200	0	0	500	60
Test6	trans_120s		5	200	0	0	500	120
Test7	trans_180s	32,400	5	200	0	0	500	180
Test8	trans_240s		5	200	0	0	500	240
Test9	trans_300s	43,200	5	200	0	0	500	300
Test10	req_250ms		5	200	0	0	250	180
Test11	req_500ms		5	200	0	0	500	180
Test12	req_1000ms		5	200	0	0	1000	180
Test13	req_2000ms		5	200	0	0	2000	180
Test14	A1_200−A3_100		5	200	0	100	500	180
Test15	A2_200		5	0	200	0	500	180
Test16	A2_200−A3_100		5	0	200	100	500	180
Test17	A1_100−A2_100−A3_100		5	100	100	100	500	180
Test18	A3_200		5	0	0	200	500	180

^*a*^ Explanations: *T*[*s*]—test duration, *R*—number of the OSTS replicas, *S*1—number of players with game strategy *A*1, *S*2—number of players with game strategy *A*2, *S*3—number of players with game strategy *A*3, *T_R_*[*ms*]—time between player requests, *T_T_*[*s*]—time between the execution of the transaction.

## Data Availability

New data were created and analyzed in this study. These data can be accessed here: https://github.com/trak2023z/Stock, accessed on 18 December 2022.

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
