# Peer review of "Knowledge Extraction and Discovery about Web System Based on the Benchmark Application of Online Stock Trading System"

_sensors, 2023, doi:10.3390/s23042274_

Round 1

Reviewer 1 Report

Advantages:

The topic of this article is very interesting and meaningful. However, the research in this article is in-depth.

This is a good analytical report with practical experiments implementation. 

The design of manuscript is well structured.

Introduction and conclusion parts are acceptable.

Background and literature review (related works) is very valuable.

The experimental part is very impressive.

Disadvantages:

I hope that the manuscript corresponds to the topic of the journal “Sensors”.

The manuscript leader T.Rak has a 5 self-citations in the reference list- isn't it too much? If MDPI journal policy allows it, then no objection.

Simple comments:

-          Line 113. Maybe use Debnath. Biplob is name.

-          Line 487. Year of publication?

In my opinion, experimental part (4. Analysis of requests traffic) is the best part in this manuscript – logical sequence of research results is given. But it feels like the part is too long. If other reviewers also have objections to it, then it is worth thinking about splitting it into two parts.

Author Response

Dear Reviewer,

Thank you very much for your review and valuable comments. You may find our responses to your comments below. All changes were marked in the red font.

Question/Comment 1:

Line 113. Maybe use Debnath. Biplob is name.

Answer 1:

Thank you for your comment. We have corrected the paper.

Question/Comment 2:

-          Line 487. Year of publication?

Answer 2:

Added.

Question/Comment 3:

In my opinion, experimental part (4. Analysis of requests traffic) is the best part in this manuscript – logical sequence of research results is given. But it feels like the part is too long. If other reviewers also have objections to it, then it is worth thinking about splitting it into two parts.

Answer 3:

Yes, this part is long indeed, however we think about future works and we investigate all possible cases.

Thank you once again.

Best regards,

Authors

Reviewer 2 Report

l.211-218 Timestamp information is missing from the majority of these requests.

Figure 2: Again, timestamps are excluded from the flow, which makes the algorithm non-deterministic. What is there is an offer and two buying bids with identical characteristics? Who ends up being the actual buyer? What if there are 2 consecutive requests from the same buyer for the same position happening before the seller responds?

Fig.3-4 If I am reading this correctly, the case of a company changing offer price is not covered (which could happen between one bid and the next).

Table 2: Why is T_T[s] is measured in seconds? The numbers seems very high.

Fig.9-10 What type of a CPU was used for the tests? How many cores? Why is the memory size stated as 20G and 30G (these are not configurations that one can buy, it can be 8G (very rare), 16G, 32G, 64G, and on).
Please describe the HW setup of a server used for these tests properly.

l.386-387 If the purpose of this system is to study realistic conditions, then there are two different setups - one of human users where these intervals are irrelevant, and one for automatic trading apps that are as fast or even faster than that - please cover this issue in your background section.

General: implementation details are missing, such as programming language, interface details, code availability and so on.

Author Response

Dear Reviewer,

Thank you very much for your review and comments. You may find our responses to your comments below. All changes were marked by a red font.

Question/Comment 1:

l.211-218 Timestamp information is missing from the majority of these requests.

Answer 1:

The loader (APP2) saves the timestamp in its database at the time of sending the request to the stock exchange (APP1). Only when generator (APP2) received a response from the stock exchange (APP1), it calculates how long the request took. The stock exchange database don’t have request timestamps. Look at: “Data Availability Statement: New data were created and analyzed in this study. This data can be found here: https://github.com/trak2023z/Stock.”.

Example fragment data log:

timestamp,apiTime,applicationTime,databaseTime,endpointUrl,queueSizeForward,queueSizeBack,replicaId

1663098851793,494,252,104,do-register,0,0,afa6a754-d407-4b30-886a-a31cece611f9

1663098851825,517,241,95,do-register,0,0,069d6e61-d0f2-4bae-96ba-7231c63fc2ab

1663098852050,499,136,5,do-register,0,0,afa6a754-d407-4b30-886a-a31cece611f9

1663098852071,495,153,13,do-register,0,0,069d6e61-d0f2-4bae-96ba-7231c63fc2ab

Question/Comment 2.1:

Figure 2: Again, timestamps are excluded from the flow, which makes the algorithm non-deterministic.

Answer 2.1:

As it was mentioned earlier timestamp is included in the calculations.

Question/Comment 2.2:

What is there is an offer and two buying bids with identical characteristics? Who ends up being the actual buyer? What if there are 2 consecutive requests from the same buyer for the same position happening before the seller responds?

Answer 2.2:

We have two types of offers: sell and buy. In transactions, we are looking/putting together corresponding offers from one type and the second type from each other.

During the transaction, the algorithm retrieved buy and sell offers for each company. The offers were sorted from the highest offer amount to the lowest offer, the sale offers from the lowest offer amount to the highest offer, and the offer that was saved first in the database was processed first. The first offer is included in the transaction.

Question/Comment 3:

Fig.3-4 If I am reading this correctly, the case of a company changing offer price is not covered (which could happen between one bid and the next).

Answer 3:

The change in the stock price occurs after all transactions for a given company have been completed. The price selection in the algorithms is based on this rate. We added it to the diagram in Fig. 2.

If any offers have been processed, the share price changes.

Question/Comment 4:

Table 2: Why is T_T[s] is measured in seconds? The numbers seems very high.

Answer 4:

This is the time between the offer processing period (called transactions). Transactions on offers are not performed continuously, only from time to time. It is used in tests. As mentioned in 379. – 381.: “The too-low and too-high values of these delay cause problems with the functioning of the application, which has been checked.” If we try to reduce this interval of time we had a problem with the performance of our system. In this case, our tests would be biased.

Question/Comment 5:

Fig.9-10 What type of a CPU was used for the tests? How many cores? Why is the memory size stated as 20G and 30G (these are not configurations that one can buy, it can be 8G (very rare), 16G, 32G, 64G, and on).

Please describe the HW setup of a server used for these tests properly.

Answer 5:

As we wrote: "We deploy a web application in a container in the cloud.". The memory size was chosen experimentally because of the load, which must always be less than 100 %.

We added to “3.1. Hardware Architecture” subsection:

All parts of the system were implemented based on a PIONIER cloud environment (https://cloud.pionier.net.pl/) with Intel Xeon E312xx (Sandy Bridge) processors. The sample memory size was selected experimentally for request processing performance lower than 100% in all cases.

Question/Comment 6:

l.386-387 If the purpose of this system is to study realistic conditions, then there are two different setups - one of human users where these intervals are irrelevant, and one for automatic trading apps that are as fast or even faster than that - please cover this issue in your background section.

Answer 6:

The background about “think time” was added in “Related works” section:

The load generators [13,14] that define web workloads imitate the behavior of thousands of concurrent users in a web browser. Existing generators mostly use different distributions for representing the time between requests (client think time) [15].

Question/Comment 7:

General: implementation details are missing, such as programming language, interface details, code availability and so on.

Answer 6:

The software used to prepare a benchmark was described in "3.2. Software Architecture" subsection. It is Java Spring Framework, RabbitMQ message broker, and Docker Swarm containers. Also, full implementation code you can find on Github – links in 3.2 subsection.

Thank you once again.

Best regards,

Authors

Reviewer 3 Report

Authors are suggested to reduce the size of the conclusion and avoid itemization. Furthermore, authors are required to increase the font size in almost all figures to improve readability. In Figs. 2 and 3, the text must fit within the flowchart blocks.

Author Response

Dear Reviewer,

Thank you very much for your review and comments. You may find our responses to your comments below. All changes were marked by a red font.

Question/Comment 1:

Authors are suggested to reduce the size of the conclusion and avoid itemization.

Answer 1:

Thank you for your comment. We have corrected the paper. We have re-edited “Conclusions” section.

Question/Comment 2:

Furthermore, authors are required to increase the font size in almost all figures to improve readability.

Answer 2:

Done.

Question/Comment 3:

In Figs. 2 and 3, the text must fit within the flowchart blocks.

Answer 3:

Done.

Thank you once again.

Best regards,

Authors

Round 2

Reviewer 2 Report

All my comments were adequately addressed.